# Microtubules provide directional information for core PCP function

Maja Matis[1][*][†], David A Russler-Germain[1][‡], Qie Hu[2], Claire J Tomlin[2], Jeffrey D Axelrod[1]

[1]Department of Pathology, Stanford University School of Medicine, Stanford, United States; [2]Department of Electrical Engineering and Computer Sciences, University of California, Berkeley, Berkeley, United States

**Abstract** Planar cell polarity (PCP) signaling controls the polarization of cells within the plane of an epithelium. Two molecular modules composed of Fat(Ft)/Dachsous(Ds)/Four-jointed(Fj) and a 'PCP-core' including Frizzled(Fz) and Dishevelled(Dsh) contribute to polarization of individual cells. How polarity is globally coordinated with tissue axes is unresolved. Consistent with previous results, we find that the Ft/Ds/Fj-module has an effect on a MT-cytoskeleton. Here, we provide evidence for the model that the Ft/Ds/Fj-module provides directional information to the core-module through this MT organizing function. We show Ft/Ds/Fj-dependent initial polarization of the apical MT-cytoskeleton prior to global alignment of the core-module, reveal that the anchoring of apical non-centrosomal MTs at apical junctions is polarized, observe that directional trafficking of vesicles containing Dsh depends on Ft, and demonstrate the feasibility of this model by mathematical simulation. Together, these results support the hypothesis that Ft/Ds/Fj provides a signal to orient core PCP function via MT polarization.

**\*For correspondence:** matism@uni-muenster.de

**Present address:** [†]Institute of Neurobiology and Behavioral Biology, University of Münster, Münster, Germany; [‡]Washington University School of Medicine, St. Louis, United States

**Competing interests:** The authors declare that no competing interests exist.

**Reviewing editor**: Helen McNeill, The Samuel Lunenfeld Research Institute, Canada

## Introduction

In *Drosophila* and in vertebrates, six proteins constituting a 'core' PCP module acquire asymmetric distributions to polarize epithelial cells along a planar axis (*Goodrich and Strutt, 2011*). In the fly wing epithelium, three of the six proteins, Frizzled, Dishevelled and Diego (Dgo), become enriched at the distal adherens junctions (AJ), two, Van Gogh (Vang) and Prickle (Pk) localize to the proximal side, while Starry night/Flamingo (Fmi) localizes to both proximal and distal sides (*Axelrod, 2009*). Preferential interactions between Fmi/Fz and Fmi/Vang complexes across cell boundaries (*Lawrence et al., 2004*; *Chen et al., 2008*; *Strutt and Strutt, 2008*) and intercellular feedback loops (*Tree et al., 2002*; *Amonlirdviman et al., 2005*) can account for intracellular segregation of these complexes and coordinated alignment among neighboring cells. However, it remains unclear how this local polarity is globally oriented with respect to the tissue axes.

It is proposed that the Ft/Ds/Fj system, comprising the atypical cadherins Ft (*Yang et al., 2002*), Ds (*Adler et al., 1998*) and the Golgi-resident protein Fj (*Zeidler et al., 1999*), acts as a 'global' PCP module, transducing tissue level directional cues encoded by opposing Ds and Fj expression gradients, to orient the core PCP module (*Yang et al., 2002*; *Ma et al., 2003*). Though a mechanism that might transmit a directional signal from the Ft/Ds/Fj module to the core module is suggested by existing observations, important additional data are needed to support the model.

In the *Drosophila* pupal wing, apical non-centrosomal MTs are aligned along the proximal distal axis prior to the onset of hair growth (*Fristrom and Fristrom, 1975*; *Eaton et al., 1996*; *Turner and Adler, 1998*; *Shimada et al., 2006*). The Ft/Ds/Fj module plays an incompletely defined role in organization of these MTs (*Harumoto et al., 2010*), and MT-associated vesicles containing Fz are observed to preferentially move in a plus-end directed fashion toward the distal cell cortex (*Shimada et al., 2006*),

**eLife digest** Almost all cells exhibit some sort of polarity: the epithelial cells that line the digestive tract, for example, have an apical domain, which faces out, and a basal domain, which faces the tissue underneath. Some epithelial cells also exhibit planar cell polarity: this involves key structures within the cell being oriented along an axis within the plane of an epithelium. Disruption of planar cell polarity is associated with various developmental defects.

It is known that the planar polarity of epithelial cells relies on two molecular complexes—a 'core' complex and a signaling complex called the Ft/Ds/Fj system—working together. While each of these complexes contributes to whole tissues having the correct polarity, the way they interact to achieve this is not fully understood.

Now, by studying epithelial cells in the wings of fruit flies, Matis et al. have provided evidence for a specific model for this interaction. The process starts with the Ft/Ds/Fj signaling complex, which orients structures called microtubules inside the cell. Microtubules are involved in providing structural support for cells, and also in the transport of organelles within cells.

Once the microtubules are oriented in the correct direction, they help to orient the core complex by moving some of the proteins that make up this complex in a specified direction. An important future challenge will be to understand how the proteins in the Ft/Ds/Fj system interact with microtubules to give them their orientation.

leading to the hypothesis that Ft/Ds/Fj signals via these MTs to orient core PCP function. However, a comprehensive spatiotemporal correlation between Ds and Fj gradients, MT orientation and direction of core protein polarization has not been examined, nor have corresponding effects of global Ft/Ds/Fj perturbations on MTs and directional vesicle trafficking been examined.

In this study, we provide additional evidence for this model in the *Drosophila* wing. We find that the apical microtubule (MT) cytoskeleton (*Eaton et al., 1996*; *Shimada et al., 2006*; *Harumoto et al., 2010*) shows strong spatial and temporal correlation with core protein asymmetry throughout wing development. We show that, in the developing wing, Ds and Fj signal through a PCP-specific domain of Ft, together with one or more partially redundant, additional signal(s), to polarize these apical MTs. Ft coordinates association of MTs with apical intercellular junctions, suggesting that Ft and Ds spatially regulate capture and organization of the apical MT cytoskeleton. We show that, in addition to Fz, vesicles containing Dsh are transcytosed on these MTs, and that transcytosis is disrupted in *ft* or *ds* mutant tissue, suggesting that this trafficking provides directional bias for core protein localization. Together, our results support the hypothesis that global polarity information is provided by the Ft/Ds/Fj module and other signals to orient the apical MT network, which in turn orients polarization of the core PCP module.

## Results

### Spatiotemporal correlation of MT alignment and core PCP protein polarization

Apical MT alignment and orientation of core PCP protein domains have been shown to correlate, but have only been examined in several small domains during late pupal wing development (*Shimada et al., 2006*; *Harumoto et al., 2010*) (between 14 and 30 hr after puparium formation [APF]) (See also *Figure 1—figure supplement 1B–D*). If MT alignment provides directional bias for core protein polarization, one should observe a spatiotemporal correlation across the entire wing throughout the time core PCP proteins are polarized.

Core PCP protein polarization with respect to the tissue axes is first observed during larval wing development (*Classen et al., 2005*). We therefore surveyed apical MT structure beginning in third instar. To facilitate this broad analysis, we used tubulin staining. While foregoing the ability to determine plus-end orientation as provided by analysis of EB1 comets, this approach enables analysis of vastly greater numbers of MTs than does the EB1 assay, and also provides the potential to distinguish a more stable, anchored population, though in fact we see a strong correlation between results from both methods (*Figure 1—figure supplement 2A–B*'; *Video 1*). Similarly, both anti-tubulin and anti-tyrosinated tubulin antibodies produce indistinguishable results (*Figure 1—figure supplement 2C*).

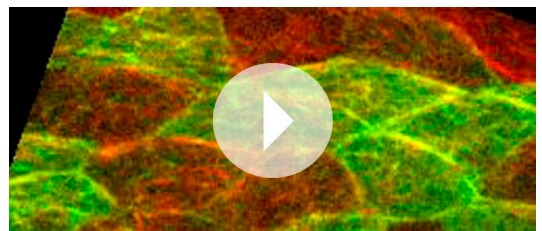

**Video 1**. Live in vivo imaging of EB1::GFP and Cherry::Jupiter. Live in vivo imaging of EB1::GFP comets and Jupiter::Cherry labeled MTs in 24 hAPF pupal wing. Refers to *Figure 1—figure supplement 2*.

The earliest evident apical tubulin staining was seen in early to mid third-instar, and revealed an asymmetrical accumulation of tubulin in mostly single 'dots' within each cell (*Figure 1A*). Image analysis of the tubulin dots revealed a significant bias of dots localizing on the proximal side of the cell (side closest to the hinge fold; *Figure 1—figure supplement 1A–A''*). Shortly thereafter, MTs appear to fan out from these sites toward the center of the wing pouch (the future distal side) (*Figure 1B*). In EM images, these early MTs appear as dense bundles anchored to proximal junctions (*Figure 1E*).

Throughout wing development, image analysis software ('Materials and methods') applied to fluorescence images demonstrated that MTs are strongly aligned along the evolving P/D axis, and the orientation of MTs was reflected in the evolving pattern of polarized PCP proteins, from the radial pattern (P/D polarity vectors from hinge fold toward center of wing pouch, resulting in concentric circles of P/D cell boundaries) evident in third instar and early pupal stage (*Figure 1C'–D'*) to the parallel pattern of 19–30 hAPF pupal wings that presages the hair polarity pattern (*Figure 1—figure supplement 1B''–D''*; *Strutt et al., 2002*; *Ma et al., 2003*; *Matakatsu and Blair, 2004*; *Classen et al., 2005*; *Rogulja et al., 2008*; *Aigouy et al., 2010*; *Hogan et al., 2011*; *Sagner et al., 2012*). Importantly, at the time the MT dots appear, there is no evident core PCP protein asymmetry, whereas core PCP asymmetry becomes globally aligned along the P/D axis in slightly older discs only after apical junction-anchored MTs appear (*Figure 1A–C*), consistent with a requirement of the apical MT cytoskeleton for core module alignment.

It is important to note that one would not expect a perfect correlation between MT orientation and orientation of core PCP proteins. The core PCP mechanism, acting through feedback loops, is expected to optimize local alignment of core PCP proteins. This influence is stronger than the directional input produced by global directional signals, and is therefore expected produce the most locally coordinated possible alignment despite the possibility of discontinuities or irregularities in the underlying global biasing inputs (*Ma et al., 2003*). Nonetheless, strong correlation was seen at all times and locations examined.

## Ultrastructure of apical MTs

Transmission Electron Microscopy (TEM) of 24 hr wings confirmed the previously described polarized organization of MTs that reach across the cell (*Eaton et al., 1996*; *Shimada et al., 2006*; *Harumoto et al., 2010*; *Figure 2A*), and also revealed that the previously observed associations of planar MTs with apical intercellular junctions (*Fristrom and Fristrom, 1975*) form juxtaposed, intercellular structures with MT anchoring sites on adjacent membranes of neighboring cells (*Figure 2A,C–E*). These anchoring sites were preferentially observed at P/D cell boundaries (*Figure 2B*). The dense, mostly single, MT organizing centers in each cell observed at the 'dot' stage (*Figure 1E*) evidently evolve into an arrangement in which multiple organizing centers are distributed around the cell in an oriented and polarized arrangement (*Figure 2*).

We wished to determine how apical MTs are captured or nucleated at membrane junctions by staining for candidate proteins. EM images showed no association of centrosomes with MTs in non-dividing cells throughout wing development, suggesting that apical MTs are nucleated elsewhere (*Figure 2F*). We detected the minus-end binding proteins γ-tubulin and Patronin (*Stearns and Kirschner, 1994*; *Goodwin and Vale, 2010*) inside the cell, but not at the cell cortex, where they have been observed in other contexts (*Meng et al., 2008*; *Feldman and Priess, 2012*; *Figure 2—figure supplement 1A*). Consistent with this, *patronin* knockdown (*Mummery-Widmer et al., 2009*) shows no PCP phenotype. MT associated proteins β-catenin (Armadillo) (*Ligon et al., 2001*; *McCartney et al., 2001*), α-catenin (*McCartney et al., 2001*) and PAR-1 (*Doerflinger et al., 2003*; *Harumoto et al., 2010*) are all present symmetrically at the cell cortex but did not show asymmetric localization at the AJs, suggesting they are not involved in apical MT anchoring (*Figure 1—figure supplement 1B,C*). These results imply that there is an alternative mechanism that nucleates early, apical non-centrosomal MTs.

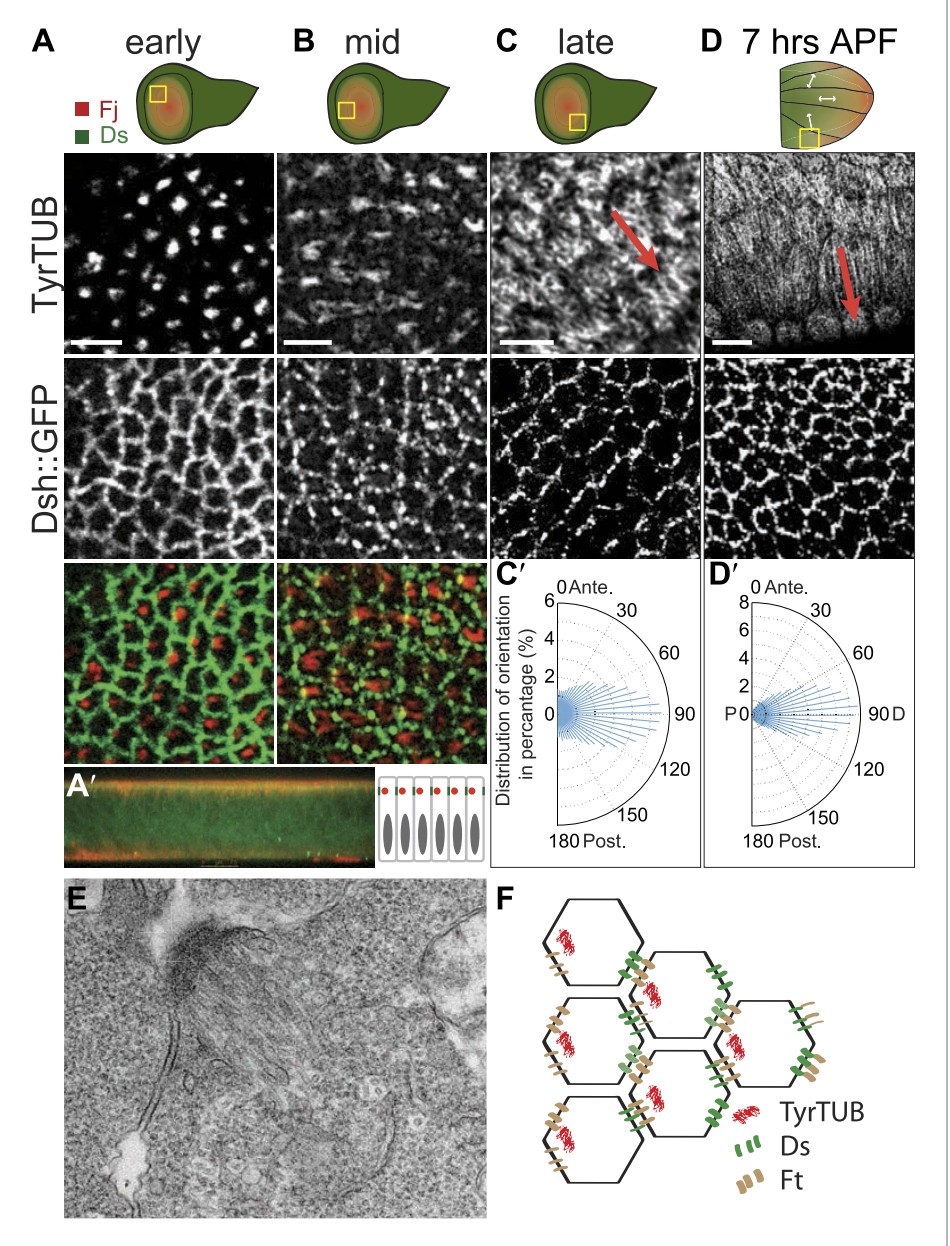

**Figure 1**. Apical junction-anchored MTs appear in early third-instar wing discs. (**A–C**) Tubulin staining in progressively older third-instar wing pouch. Note the asymmetric accumulation of tubulin on the proximal side of the cell in early third-instar wing pouch and growing MTs observed in mid-late third-instar wing pouch. White arrows in the first wing disc cartoon show the proximal distal axis, with distal in the center of the wing pouch, and proximal tissue forming a ring around the future wing blade. (**D**) MTs in 7 hAPF pupal wing. Scale bar: 5 μm. (**A'**) A vertical cross-section of the early third-instar wing disc. Apical is at the top. (**C'** and **D'**) Orientations of MTs, in the regions shown, calculated using OrientationJ, aligned with the P-D axis at each stage and location (arrows). (**E**) TEM micrograph of MTs in early third-instar wing pouch. Note anchored MTs at intercellular junction. (**F**) Schematic of organization of MTs, Ft and Ds in early third instar wing cells.

The following figure supplements are available for figure 1:

**Figure supplement 1**. Organization of the apical MT cytoskeleton in early third-instar wing discs.

**Figure supplement 2**. Correlation between EB1 comets and MT orientation in *Drosophila* wing epithelium.

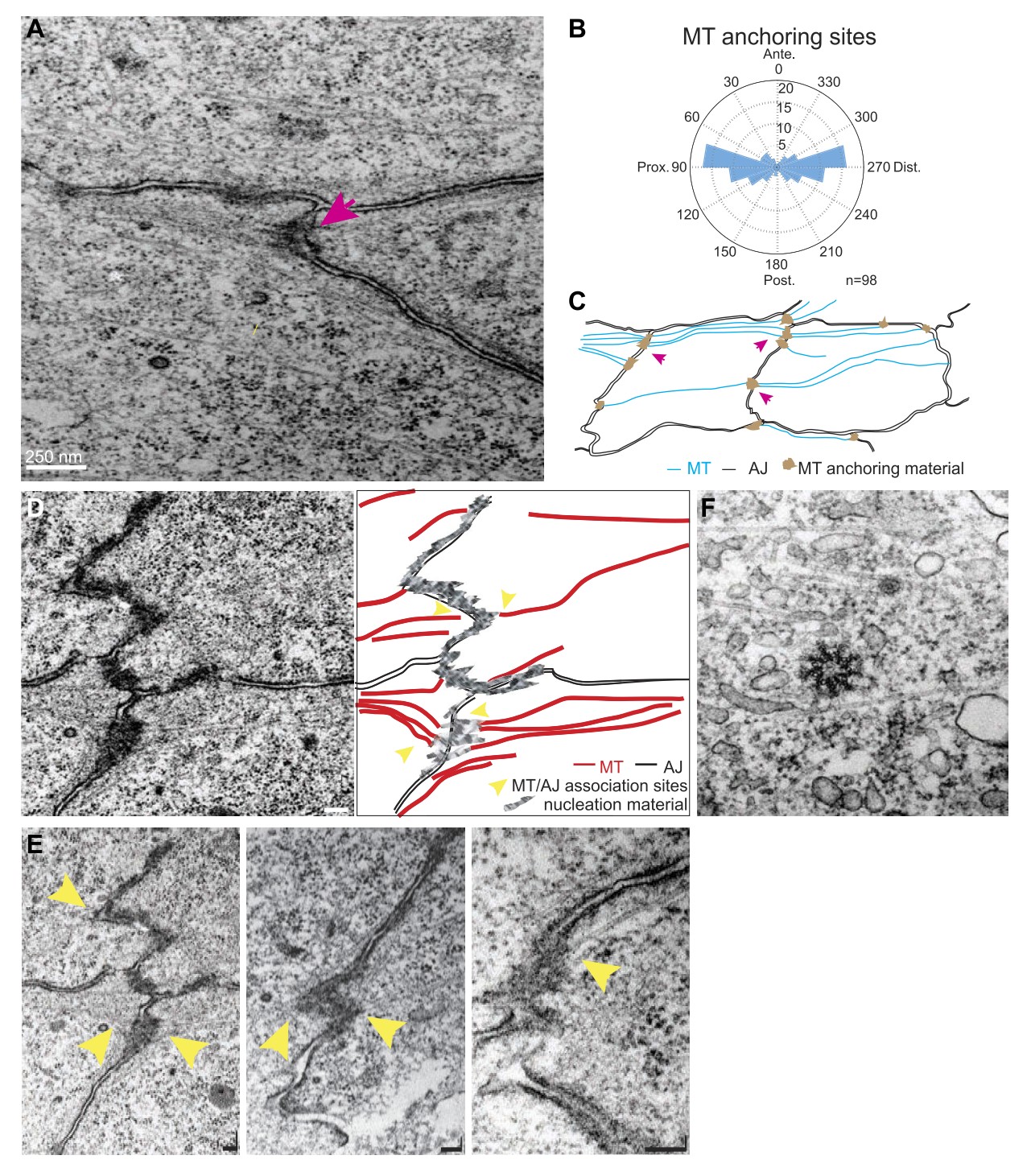

**Figure 2**. TEM micrographs showing apical MT organization in 24 hAPF pupal wings. (**A**) TEM micrograph of MTs in 24 hAPF pupal wing. Note MTs anchored at both sides of intercellular junctions (arrow). (**B**) Graph depicting the localization of MT anchoring sites in the cell relative to the cell centroid. Plot is composed of 20 bins of 18° each. (**C**) Organization of the apical MT cytoskeleton traced from a single micrograph showing MTs spanning the cell preferentially in the P-D orientation. (**D**) TEM revealing planar MTs with anchoring sites on adjacent P-D cell membranes juxtaposed between two neighboring cells, with a sketch of the junctions and MTs. (**E**) Three additional images. The first is a lighter exposure of the image in **D**, to better reveal the structure of the junctions. Yellow arrowheads show associations of planar MTs with apical intercellular junctions. Scale bars, 100 nm. (**F**) TEM micrograpf showing no association of centrosome with MTs.

*Figure 2. Continued on next page*

*Figure 2. Continued*

The following figure supplement is available for figure 2:

**Figure supplement 1**. Distributions of potential MT interacting proteins.

## Ds and Fj gradients and Ds subcellular asymmetry align with MT orientation

What is the spatial signal that polarizes non-centrosomal MTs? Prior evidence suggests that the Ft/Ds/Fj pathway plays a role in organization of apical MTs (*Harumoto et al., 2010*; *Marcinkevicius and Zallen, 2013*). In the wing, as in other tissues, Ds and Fj are expressed in opposing gradients (*Strutt et al., 2002*; *Ma et al., 2003*; *Matakatsu and Blair, 2004*; *Rogulja et al., 2008*; *Aigouy et al., 2010*; *Hogan et al., 2011*; *Sagner et al., 2012*) which are converted into subcellular asymmetries of Ft and Ds heterodimers (*Brittle et al., 2012*). Biased subcellular orientations of asymmetric Ft-Ds heterodimers could play a role in polarization of the apical MT cytoskeleton. Consistent with this, the direction of MT growth between 14 hAPF and 30 hAPF have been shown to correlate with Ds and Fj gradient direction (*Strutt et al., 2002*; *Ma et al., 2003*; *Matakatsu and Blair, 2004*; *Hogan et al., 2011*) in the central part of the pupal wing (*Harumoto et al., 2010*). The possibility that the Ft/Ds/Fj system may organize apical MTs is also supported by prior EB1 comet assays showing that apical MTs are abnormal in *ds* mutant pupal wings, and that Ds or Ft misexpression perturbs their orientation (*Harumoto et al., 2010*). However, the reported assays were too limited to draw strong conclusions about overall architecture or evolution of the MT pattern (*Harumoto et al., 2010*).

Here, our analysis shows that MTs are generally aligned with Ds and Fj gradients from their first appearance in third instar discs (*Rogulja et al., 2008*), when they emerge on proximal sides of the cell cortex (*Figure 1A*). Throughout larval wing discs and pupal wings, MT orientation correlates with the direction of Ds and Fj gradients (*Figure 1A–D*, *Figure 1—figure supplement 1*). As noted previously (*Brittle et al., 2012*), in imaginal discs, when the tissue is small and gradients appear to be steeper (*Figure 3—figure supplement 1C*), marked subcellular asymmetry of Ft localization is observed that substantially overlaps core protein distribution (*Figure 3—figure supplement 1A,B*). We detect a similar relationship in pupal wings (*Figure 3—figure supplements 2 and 3*). The steepest regions of the gradients correspond to the most polarized MTs (compare *Figure 1D* to *Figure 3—figure supplement 2A*). Therefore, Ds and Fj gradients are appropriately aligned to polarize the MT cytoskeleton and thereby bias core PCP protein polarization from its earliest appearance. These data are consistent with the temporal requirement for Ds in the larval stage (*Matakatsu and Blair, 2004*; *Aigouy et al., 2010*).

Note that caution is required in deciphering the Ds and Fj gradients. Existing data for Fj expression all derive from Fj-LacZ expression, and should therefore be considered only approximate at best. For Ds, low magnification images can be deceptive, since excess cytoplasmic signal, in contrast to the relevant membrane pool, cannot be distinguished, and because smaller cells give the appearance of higher concentrations in low magnification even if membrane intensity is constant. Therefore, we have focused on analyzing subcellular asymmetric localization of Ds in high magnification images, and examples of these data are shown in *Figure 3—figure supplements 1–4*. We observe that, for the most part, asymmetry of Ds localization is very similar to that of core protein localization.

One exception is the posterior margin of the wing, where Ds often appears to be oriented more posteriorly than is Fmi, though overall levels of asymmetry are modest (*Figure 3—figure supplement 4*, box 5). In this region, Ds and Fmi polarities therefore appear to be less tightly coupled. While we do not know the reason for this, we can speculatively suggest several possibilities. These include (1) the tendency for the core system to promote local alignment producing a more parallel arrangement than would a direct readout of the Ds pattern; (2) that oppositely oriented Ft-Ds heterodimers are unevenly distributed despite the even distribution of total Ds; (3) the tendency of MTs to align along the long axis of the cell may be stronger than the influence of Ft-Ds. Consistent with this, MT orientation correlates more strongly with the long axis of cells than with the Ds asymmetry (*Figure 3—figure supplement 4*, box 5). Finally, (4) we cannot rule out the possibility that other unknown signals may also be acting on MT orientation (see 'An independent directional signal in the wing periphery?').

## Hippo signaling-independent effects of Ft/Ds/Fj on polarity and MTs

It has been suggested that PCP defects in *ds* and *ft* mutants may be due to activation of the Hippo tumor suppressor pathway, which is also controlled by Ft/Ds/Fj, because *ft* mutant larval wing discs with rescued Hippo signaling show only weak PCP defects limited to the most proximal part of the wing despite the clonal PCP phenotypes observed in pupal and adult wings (*Brittle et al., 2012*). To better assess whether the Ft/Ds/Fj pathway modulates the MT cytoskeleton independent of Hippo signaling, and to do so across the expanse of the wing, we analyzed *ft*$^{null}$ mutant flies rescued with FtΔECDΔN-1, a truncated form of Ft lacking a PCP signaling domain (*Matakatsu and Blair, 2012*). These flies are deficient for PCP signaling, but competent for Hippo pathway regulation. They showed PCP defects in the proximal-central part of the adult wing, displaying swirling patterns reminiscent of those in *ft* clones (*Ma et al., 2003*) (*Figure 3A*; *Matakatsu and Blair, 2012*). At 24 hAPF, MTs in the proximal and central part of the wing, where hair polarity is often disturbed, were randomized (*Figure 3A,A′*, *Figure 3—figure supplement 5A,A′*). In contrast, the peripheral and distal regions of these wings had more coherent hair polarity, with hairs pointing more toward the wing margin than in wildtype wings (*Figure 3A*), mirroring the orientation of core PCP protein domains (*Figure 3—figure supplement 5A,A″*). In these peripheral regions, MTs were ordered and oriented with the hairs and core PCP domains (*Figure 3A″*). Flies in which the Ds and Fj gradients were removed showed an essentially identical phenotype (*ds*$^{38k}$ *fj*$^{N7}$/*ds*$^{UA071}$ *fj*$^{d1}$; *UAS-Ds/TubP-Gal4*; *Figure 3—figure supplement 5B–B″*). Finally, MT orientation was randomized in *ft* or *ds* clones in the same proximal part of the wing where it is disturbed in *ft* mutant wings rescued for Hippo signaling (*Figure 3—figure supplement 5C*; see also [*Ma et al., 2008*]). These data show that in the central part of pupal wings, MT orientation, core PCP protein polarity and adult polarity are strongly dependent on PCP signaling through Ft (*Figure 3—figure supplements 5 and 6*). They also suggest the existence of an additional signal, perhaps from the wing margin, that can orient MTs in the periphery of the wing.

## Ft-Ds signaling is instructive for MT orientation

To determine whether Ft-Ds signaling regulates MT orientation in wing discs, and to determine whether it is instructive or merely necessary, we studied the boundaries of *ft* clones. In cells surrounding *ft* clones in wing discs, where unopposed Ds within the clone is expected to recruit excess Ft to the neighboring cell boundaries, nascent MT bundles are inappropriately polarized toward the cell border abutting the mutant cells (*Figure 3B,B′*). Similarly, in pupal wings MTs are perpendicular to the clone boundary (*Figure 3C,C″*), consistent with the reported non-autonomy resulting from manipulating the Ft/Ds/Fj system (*Brittle et al., 2012*). To quantify this result, we applied our image analysis tool to cells bordering (n = 51) *ft* clones in regions where MTs would otherwise be expected to run parallel to the clone border. *Figure 3C″* shows that in these cells, MTs are reorganized predominantly perpendicular to the clone border. These results are consistent with the reported reversal of MT orientation in wings with an ectopic Ds gradient, although this was only examined late in the polarization process, during pupal development (*Harumoto et al., 2010*). Therefore, Ft and Ds are both required and instructive for MT organization.

## Directed Dsh vesicle trafficking depends on Ft

Distally biased microtubule (MT)-dependent trafficking of Fz-containing vesicles has been shown to occur during polarization of the core PCP proteins, and both are sensitive to MT disruption, suggesting that transport is required for polarization (*Shimada et al., 2006*). Since we have proposed that Dsh is the critical determinant that must be asymmetrically localized (*Amonlirdviman et al., 2005*; *Axelrod, 2009*), we also examined Dsh::GFP vesicle movement in developing wings in the AJ plane, between 15 and 32 hr after puparium formation (hAPF). The majority of vesicles (80%, *n* = 1192) moved along the P/D axis, and showed a significant though modest bias towards distal vs proximal transport (*Figure 4A,C–D*; *Video 2*). Dsh::GFP vesicles exhibited two distinct patterns of trafficking. First, and most commonly, vesicles emerged from one side of the cell and were transported directly across the length of the cell to be incorporated into the membrane of an opposing cellular face (*Figure 4A′*; *Video 2*). Movement was highly linear and processive, though occasional backtracking and zig-zagging was observed. Often, multiple vesicles followed similar paths in a given cell, with vesicle scission and fusion appearing to occur repeatedly at specific sites. Second, a minority of Dsh::GFP vesicles took staggered paths without directional bias, paused frequently, and often left the apical plane of the cell (*Figure 4B,B′*; *Video 3*). The former pattern likely reflects polarized

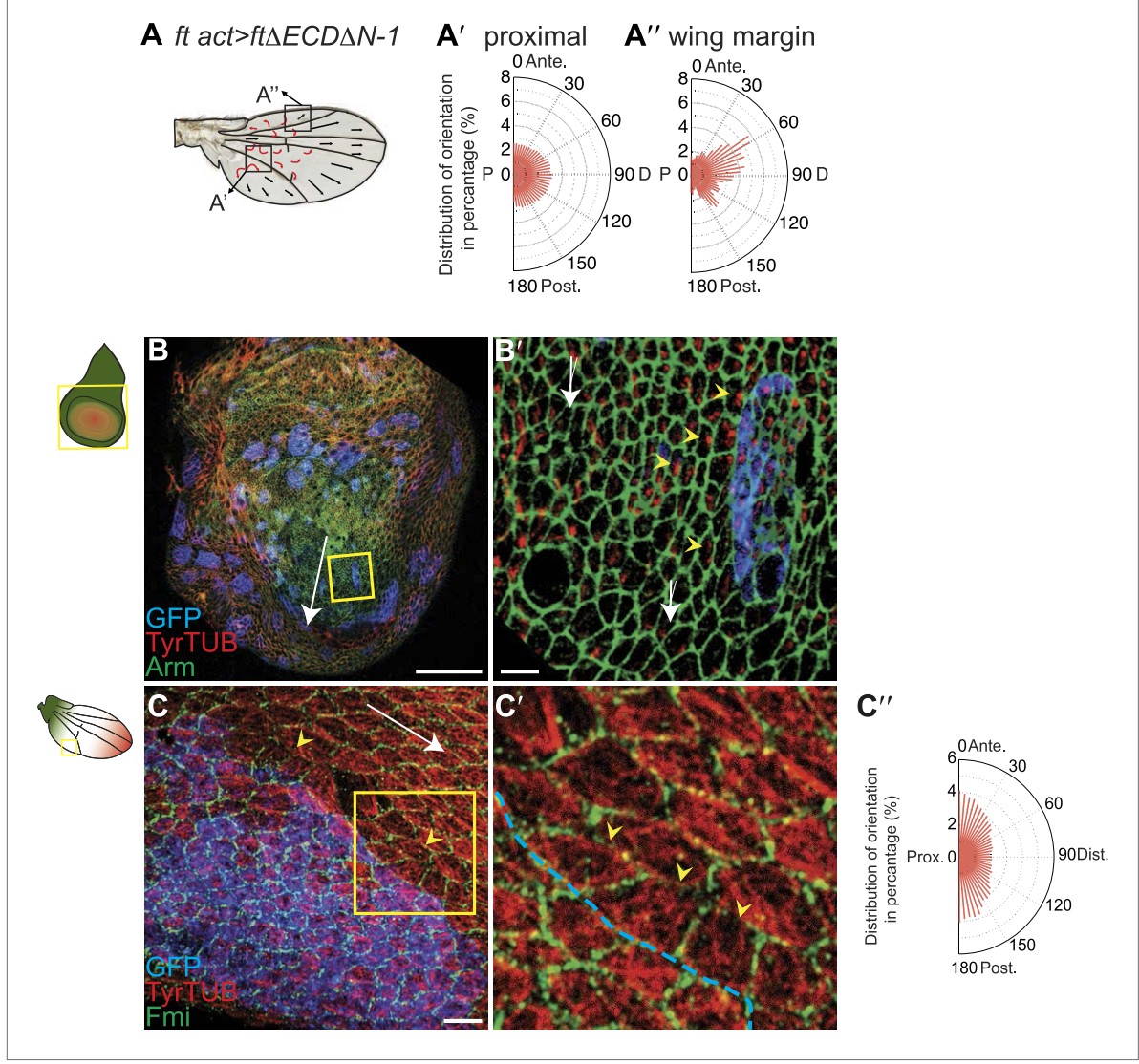

**Figure 3**. MTs are misaligned in *ft* and *ds* mutant wing. (**A**–**A″**) Analysis of hair polarity and MT orientation in *ft*[l(2) fd] / *ft*[GRV] *ActP-Gal4 UAS-FtΔECDΔN-1* mutant flies. (**A′**) MTs in the proximal and central part of the wing, where hair polarity is disturbed, are randomized. (**A″**) MTs in the distal/peripheral part of the wing are oriented with the hairs pointing toward the margin (all plots are composed of 36 bins of 5° each). (**B** and **C**) Ft-dependent instructive re-organization of the MT cytoskeleton. Wildtype cells around *ft* clones only see Ds in their mutant neighbors, and therefore preferentially accumulate Ft at their junction with the mutant cell. The orientation of MT dots is changed in wildtype cells around *ft* clones in third instar (**B** and **B′**) and 24 hr pupal wings (**C** and **C′**). White arrows show P-D axis and yellow arrowheads show MT spots initiating toward the clone rather than proximally (**B′**) or MTs organized perpendicular to the clone boundary (**C′**) rather than proximally-distally (right panels are high magnification images of left panel boxes). (**C″**) Orientations of MTs in the wildtype cells bordering *ft* clones calculated using OrientationJ. Scale bars: 50 μm (**B** and **C**) and 5 μm (**B′**–**C′**). The P-D axis is defined as a radius from the center of the wing disc to the circumference, as described in *Figure 1*.

The following figure supplements are available for figure 3:

**Figure supplement 1**. Orientations of Ft, Ds and core proteins correlate in wing disc.

**Figure supplement 2**. Orientations of Ds gradient and core proteins correlate in early pupal wing (7 hAPF).

**Figure supplement 3**. Orientations of Ds gradient and core proteins correlate in late pupal wing (24 hAPF).

**Figure supplement 4**. Orientations of Ds gradient, MTs and core proteins correlate in late pupal wing.

*Figure 3. Continued on next page*

*Figure 3. Continued*

**Figure supplement 5**. MTs in *ft* mutant wings are randomized.

**Figure supplement 6**. Wing hair polarity in *ft* mutant wings reflects core PCP orientation in these wings.

**Figure supplement 7**. Overexpression of Wnt4 affects MT polarity.

transcytosis, resulting in net transport of Dsh to the distal membrane, while the latter reflects a recycling pathway. Consistently, in fixed specimens, only a minor fraction of Dsh vesicles co-stains with the early endosome marker Rab5 or exocyst protein Sec5 (*Figure 4—figure supplement 1A*). Thus, a minority of vesicles moves through the recycling pathway while the majority of Dsh vesicles appears to be part of a transcytosis pathway. In contrast, we see no directionally biased trafficking of Vang::YFP vesicles (*Figure 4E*). Note that biased directional transport of any one component of the core PCP proteins should be sufficient to provide an input bias; bulk transport to achieve the remainder of asymmetric localization is expected to occur by diffusion in combination with feedback at intercellular junctions. Together with prior data, our observations suggest that the 'distal' components Fz and Dsh, but not 'proximal' components, are subject to directional trafficking.

The observed spatiotemporal organization of the MT cytoskeleton (*Figure 2C*) is suitable for directing biased transport of Dsh and Fz vesicles across cells (i.e., transcytosis) that could bias the direction of core PCP protein polarization. Furthermore, the repeated scission of vesicles from specific regions suggests that vesicle formation may be coupled to the locations of MT-associated junctional structures observed in EM images. Furthermore, the common directionality of these temporally clustered vesicular trafficking events suggests that individual MTs nucleated at or associated with a given junctional density, are likely to have the same polarity. Though the hypothesis that biased trafficking of Dsh and Fz depends on polarized MTs that are organized by Ft/Ds/Fj is appealing (*Harumoto et al., 2010*), no data directly link Ft/Ds/Fj function to directed vesicle trafficking. To test this, we examined Dsh::GFP vesicle movement in proximal *ft* mutant pupal wing tissue. We observed that, in comparison to wildtype cells, vesicles moved much shorter distances (or showed no net movement), without directional bias, and lacked processivity, instead taking random paths with frequent direction changes (*Figure 4F*).

## Polarized Dsh[1] vesicle trafficking is independent of core PCP polarity

Apical MTs are oriented independent of core PCP mutants *fz* (*Shimada et al., 2006*; *Harumoto et al., 2010*), *vang* (*Harumoto et al., 2010*) and *dsh[1]* (*Figure 4—figure supplement 1B*). However, Fz vesicle trafficking was not scored in a core mutant background because vesicle production depends on most or all core proteins (*Shimada et al., 2006*). To verify that directed vesicle trafficking depends on Ft/Ds/Fj but not on core protein asymmetry, we measured movement of Dsh[1]::GFP in *dsh[1]* flies. No core protein asymmetry is evident in *dsh[1]* wings, but Dsh[1]::GFP vesicles are still produced, albeit at a lower frequency than in wildtype. We found that, unlike in *ft* mutant cells, most Dsh[1]::GFP vesicles move along the P-D axis (*Figure 4G*; *Video 4*), consistent with the presence of oriented MTs, and indicating that oriented trafficking does not depend on core protein asymmetry. We observed that Dsh[1]::GFP vesicles frequently fail to fuse with membranes, and compared to wildtype, more frequently disappear from the apical plane or do not exhibit overall net movement, perhaps reflecting defects specific to the Dsh[1] allele. These results show that apical planar MTs that direct biased transcytosis of Dsh depend on the Ft/Ds/Fj pathway, but not on core module function. Furthermore, the movement of Dsh[1] vesicles, the faster kinetics of transcytosing Dsh vesicles, and the greater processivity, compared to Fz vesicles all suggest that Dsh and Fz vesicles may be at least partially distinct populations.

## Polarization is predicted to be insensitive to the shape of the Ds and Fj gradients

Our results thus far suggest that gradients of Ds and Fj expression, by producing asymmetric orientation of Ft-Ds heterodimers, provide directional information to bias core protein polarization. Given the apparent variation in asymmetry of Ft-Ds dimers at different times and places in wing development, we wished to assess the potential consequences of this variation on core PCP protein asymmetry. We therefore simulated this mechanism by adapting our previously described mathematical model for

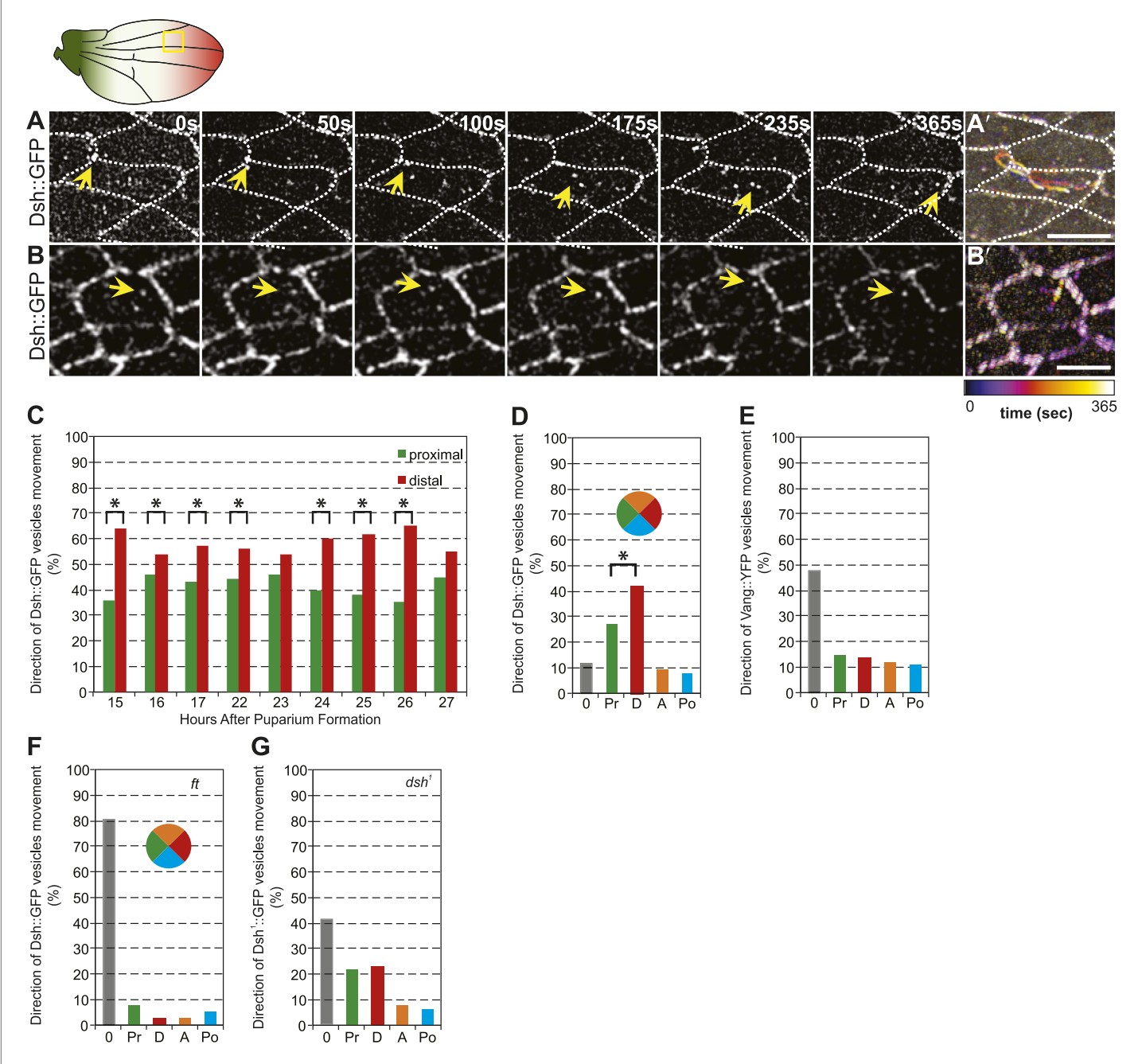

**Figure 4**. Dsh::GFP vesicles move preferentially toward the distal side of the cell. (**A** and **B**) Timelapse images of Dsh::GFP vesicles in 24 hAPF pupal wing. The majority of Dsh::GFP vesicles displayed directed, fast transcytotic movement with a distal bias (**A**). A minority of vesicles moved slowly along irregular paths (**B**) (derived from **Video 2** and **Video 3**). (**A'** and **B'**) Overlays of 73 frames from timelapse videos in **A** and **B**. Proximal is to the left and anterior is at the top. (**C**) The ratios of transcytosing Dsh::GFP vesicles at different pupal ages (n ≥ 1192). (**D**) Ratio of Dsh::GFP vesicles moving toward proximal, distal, anterior or posterior, or not moving (n = 50). (**E**) Similar plot of Vang::YFP vesicles showing absence of directed trafficking (n = 81). (**F**) Dsh::GFP vesicle movement inside *ft* clones showing little net movement and no significant bias (n = 42). (**G**) Dsh[1]::GFP vesicles in *dsh[1]* wings showing a bias to P-D vesicle movement among vesicles showing net movement (n = 120). Numbers were too small to test significance of a possible difference between P and D. Scale bars, 5 µm. Significant differences between proximal and distal at p ≤ 0.05 using the binomial test are marked with *.

The following figure supplement is available for figure 4:

**Figure supplement 1**. Vesicle identification and tubulin staining.

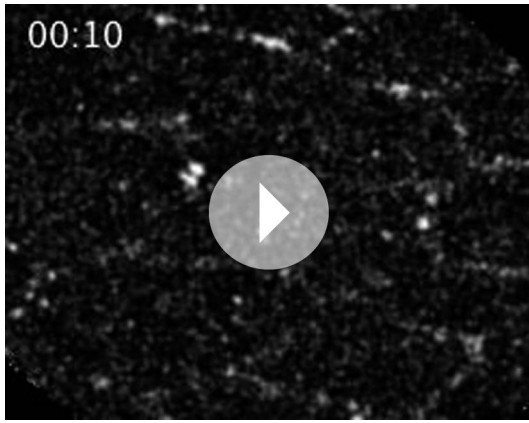

**Video 2**. Live in vivo imaging of Dsh::GFP. Live in vivo imaging of Dsh::GFP vesicles in 24 hAPF pupal wing. Refers to *Figure 4A*.

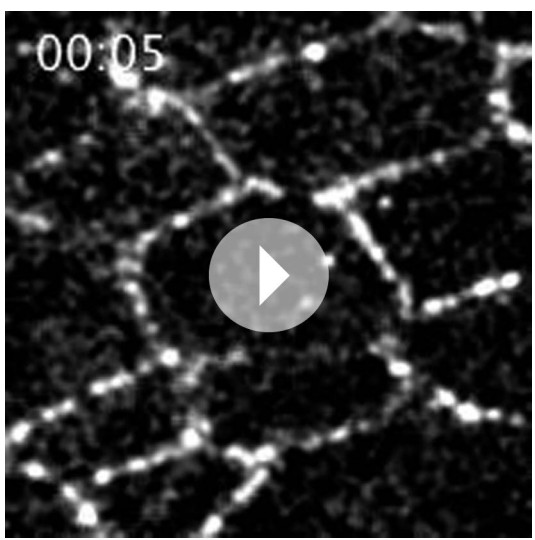

**Video 3**. Live in vivo imaging of Dsh::GFP. Live in vivo imaging of Dsh::GFP vesicles in 24 hAPF pupal wing. Refers to *Figure 4B*.

PCP signaling (*Amonlirdviman et al., 2005*; *Ma et al., 2008*). The modified model establishes a MT network with polarity determined by the relative concentrations of Ft on any side of a cell. User-defined input gradients of Ds and Fj determine Ft concentrations in a manner consistent with the experimentally defined model. Dsh is then transported toward the plus ends of MTs, while still permitting bulk movement of all components by diffusion (see *Supplementary file 1*). We first validated the model by correctly reproducing the domineering non-autonomy (or lack thereof) surrounding clones of core PCP mutants (*Figure 5—figure supplement 1*). Furthermore, we confirmed that the model correctly simulates the ability of the core module to propagate polarization across small *ft* mutant clones (*Figure 5—figure supplement 1*).

The model then allowed us to predict the response to different configurations of the Ds or Fj gradients. As discussed above, there is considerable ambiguity about the shape of the Ds gradient through wing development, but to a first approximation, it appears to undergo considerable change from larval wing discs, where there is a comparatively linear gradient, at least in the distal portion of the wing not hidden by tissue folds, to one with a steep drop and very shallow or flat portion in the 24–30 hr pupal wing (*Figure 3—figure supplements 1–4*; *Ma et al., 2003*; *Matakatsu and Blair, 2004*; *Hogan et al., 2011*). In third instar discs, gradients of Ds and Fj are roughly linear (*Figure 3—figure supplement 1C*). In simulation, oppositely oriented linear gradients of Ds and Fj polarize a field of cells with similar kinetics and identical steady state levels of polarization across the entire field. Whereas in the larval wing disc, the Ds gradient may be gradual, in the pupal wing, the gradient of Ds approaches a step gradient as it rearranges first to a projection of high Ds in the central part of the pupal wing, and later to a very high proximal

concentrations and a shallow or even flat distal distribution. Notably, simulation of a linear gradient, a step gradient, or a steep proximal Ds gradient and a shallow or flat distal gradient produces identical levels of steady state polarization across the field and similar proximal and distal kinetics, showing that the mechanism is not expected to be very sensitive to the precise shape of the Ds gradient (*Figure 5*). In the cases of a steep local Ds gradient, propagation of Ft-Ds polarity into an adjacent shallow or flat region is weak, and limited to two columns of cells (data not shown), most likely due to absence of a robust feedback mechanism in our model. Similarly, propagation of Ft-Ds polarization in vivo is seen to be much weaker than that of the core PCP mechanism (c.f. *Ambegaonkar et al., 2012*; *Brittle et al., 2012*). Therefore, propagation of polarity through the shallow or flat region is primarily due to polarization and propagation of the core PCP system.

In the distal part of the wing, MTs are oriented in the P-D direction, but have no detectable polarity bias. We therefore simulated several additional conditions. First, we simulated a steep proximal gradient with a distal zero level of Ds that produces randomized distal MT orientation in our model. Global directional input is therefore restricted to the proximal region. In this case, we see

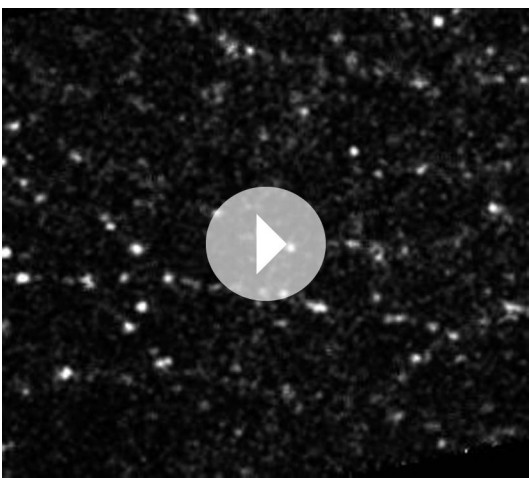

**Video 4**. Live in vivo imaging of Dsh1::GFP in dsh1 mutant flies. Live in vivo imaging of Dsh1::GFP vesicles in 24 hAPF dsh1 pupal wing. Refers to **Figure 4G**.

equivalent proximal and distal steady state polarization, but with a substantial delay in reaching steady state in the distal region, reflecting time needed to propagate polarity from proximal to distal through the core mechanism. To simulate MTs that are P-D oriented, but without a measurable plus-end bias, as are observed in pupal wings, we enforced this MT architecture in the distal wing, with a steep proximal Ds gradient. Simulation of this condition predicts modestly faster core PCP polarization compared to random distal MTs, but not to change steady state core PCP polarization. Therefore, the observed unbiased but oriented MTs in the distal wing might facilitate more rapid core PCP polarization and also maintenance of polarity in the face of perturbations.

In similar simulations, we tested the response to different configurations of the Fj gradient. Again, we found that steady state polarization is insensitive to the gradient configuration, with only slight differences in kinetics (data not shown).

A similar result was obtained when simultaneously altering the shapes of both gradients (data not shown). From these simulations, we conclude that so long as the Ds and Fj gradients are in the proper direction, their potentially evolving profiles are not expected to have a substantial effect on the resulting core PCP polarity.

## An independent directional signal in the wing periphery?

Distal (peripheral) polarity is independent of Ft function; polarity in this region may depend on a spatial signal perhaps originating from the wing margin. A candidate for this signal is the proposed redundant functions of Wnt4 and Wg. Both are expressed at the wing margin, combined loss-of-function produces a mild polarity phenotype, and overexpression of Wnt4 to a much greater extent than Wg perturbs hair polarity (*Lawrence et al., 2002*; *Lim et al., 2005*; *Wu et al., 2013*). Based on a cell culture assay, these Wnts were suggested to impact core PCP function by blocking interactions between Fz and Vang. However, our observation that a Ft independent signal might polarize MTs near the wing margin suggests that other possible mechanisms should be considered. Notably Wnt4 (but not Wg) overexpression reorganizes MTs (*Figure 3—figure supplement 7*), suggesting that a different possible mechanism for Wnt4 function should be entertained.

## Discussion

Together, our findings support the hypothesis that a polarized MT cytoskeleton orients PCP throughout wing development by directing the trafficking of Dsh containing vesicles. Furthermore, they confirm that the Ft/Ds/Fj PCP module directs orientation of this apical MT cytoskeleton, at least in the proximal central portion of the wing.

We infer that a second signal, acting near the wing margin and perhaps originating from the margin, can also organize MTs to orient the core PCP mechanism. The recent finding that Wnts expressed at the wing margin regulate PCP suggests a possible identity for this signal (*Wu et al., 2013*). We propose that in third instar wings, when the tissue is smaller, the two signals are largely redundant, so that defects in Hippo-rescued *ft* mutants are limited to the most proximal regions (*Feng and Irvine, 2007*; *Brittle et al., 2012*; *Matakatsu and Blair, 2012*; *Pan et al., 2013*), whereas in larger pupal and adult wings, Hippo-rescued *ft* mutants show larger regions of disturbed polarity. One difficulty in understanding how a wing margin-based signal might contribute to polarization is that much of the anterior and posterior margin is parallel to the direction of polarization, while the distal portion of the margin is perpendicular. Additional studies will be needed to understand potential signals from the margin.

Notably, even in regions well polarized by the presumed wing margin signal, ectopic Ds expression can reorganize polarity (*Matakatsu and Blair, 2004*; *Harumoto et al., 2010*). Furthermore, the

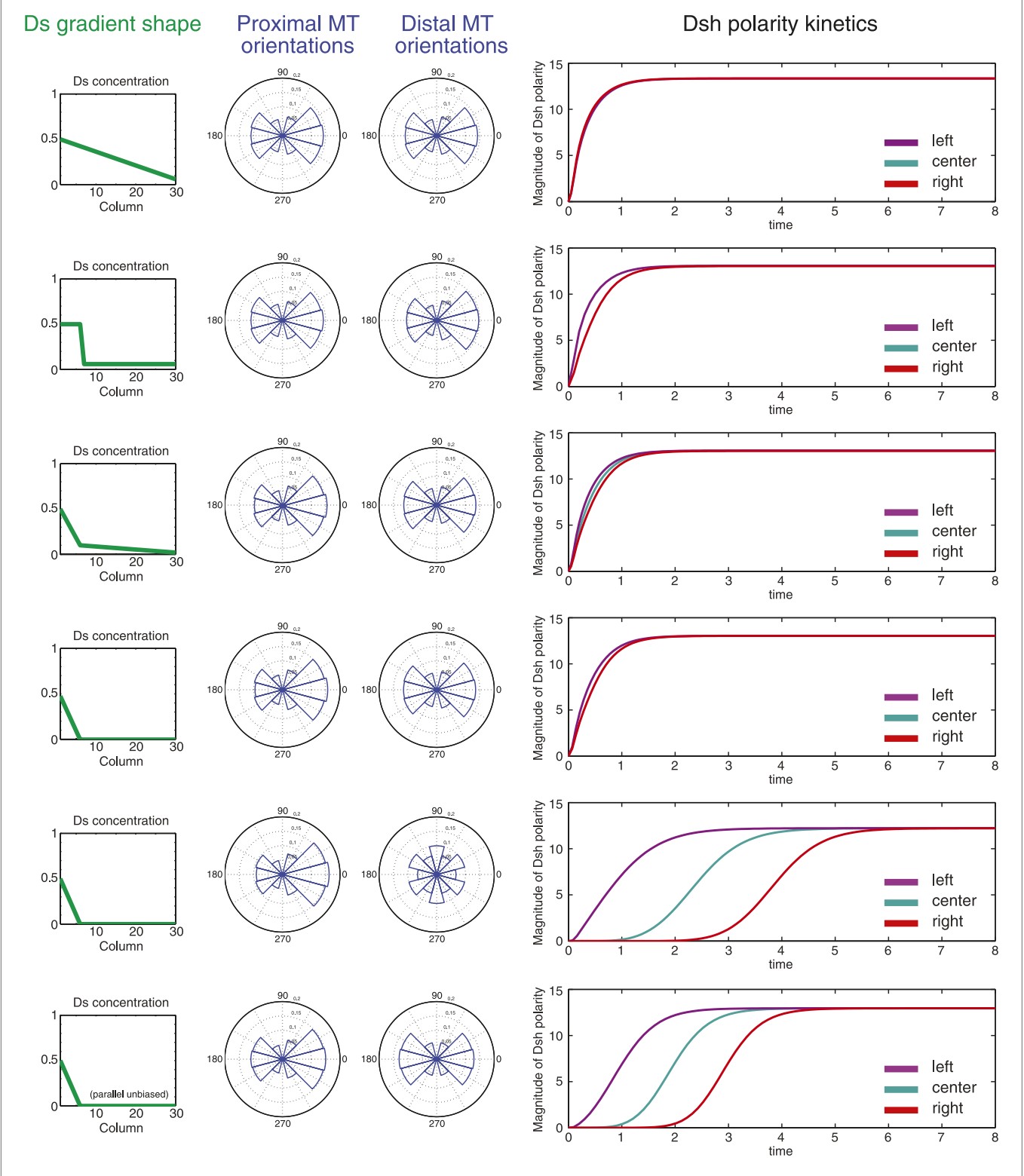

**Figure 5**. Simulations of polarization kinetics with different shapes of Ds gradients or imposed MT structures, using a mathematical model incorporating a simple representation of the Ft/Ds/Fj system to polarize MTs. Input Ds gradients are shown on the left. Resulting (or imposed, on the right side of the last example) MT organization on the proximal or distal portion of the gradient are plotted (center). Kinetics of polarization of cells in column 8 (proximal = left), column 15 (center) or column 23 (distal = right; see *Figure 5—figure supplement 1*).

*Figure 5. Continued on next page*

*Figure 5. Continued*

The following figure supplement is available for figure 5:

**Figure supplement 1**. Simulation results.

strong correspondence of MT orientation and core protein orientation throughout wing morphogenesis, their overall correspondence to Ds-Fj gradients, and the ability of altered Ft or Ds expression patterns to reconfigure MT orientation, suggest that these gradients provide instructional information for core PCP orientation, at least in the proximal and central region of the wing. This signal likely acts in conjunction with other molecular signals, particularly in the peripheral region of the wing, and perhaps with mechanical inputs such as cell flow and cell elongation (*Aigouy et al., 2010*).

Recently, it was shown that the tissue and compartment specific expression predominance of the Pk vs Spiny-legs isoforms of Pk determines the direction of Ds and Fj gradient interpretation (*Ayukawa et al., 2014*; *Olofsson and Axelrod, 2014*). Thus, for example, polarization of the core module can occur in the same direction in the Anterior and Posterior compartments of the abdomen despite oppositely oriented Ds and Fj gradients in these compartments.

In summary, we provide evidence favoring the model that the Ft/Ds/Fj global PCP module, together with a partially redundant and as yet unidentified peripheral wing signal, orients apical polarized microtubules, directing Dsh-vesicle transcytosis, and thereby imparting directional information to the core PCP module. To what extent other global signals may function in other tissues remains to be determined. The presence of polarized MTs suggests that a MT dependent global cue may also function in vertebrate PCP (*Vladar et al., 2012*).

## Materials and methods

### *Drosophila* stocks
The following fly lines and mutant alleles were used:

> *OreR,*
> *Dsh::GFP,*
> *actinP-Vang::EYFP,*
> *Ds::GFP* (*Brittle et al., 2012*),
> *ciGal4/UAS-EB1::GFP,*
> *ubi-Jupiter::mCherry,*
> *armP-Fz::GFP,*
> $ft^{l(2)\ fd}$ *FRT40A Dsh::GFP/NLS::mRFP FRT40A;* T155Gal4 *UAS-FLP/+,*
> $ft^{GRV}$ *FRT40A,*
> $ft^{GRV}$ *FRT40A/$ft^{l(2)\ fd}$ FRT40A; UAS-FtΔECDΔN1* (*Matakatsu and Blair, 2012*)/*ActP-Gal4,*
> $ds^{38k}$ $fj^{N7}$/$ds^{UA071}$ $fj^{d1}$; *UAS-Ds/TubP-Gal4,*
> *hs-FLP;* $ds^{UA071}$ *FRT40A/FRT40A Tub-Gal80; TubP-Gal4 UAS-mCD8GFP/+,*
> $ft^{Gr-V}$ *FRT40A/FRT40A Tub-Gal80; TubP-Gal4 UAS-mCD8GFP/+,*
> $ft^{l(2)fd}$ $d^{GC13}$/$ft^{l(2)fd}$ $d^1$,
> *enGal4/UAS-Wnt4.*

### Immunohistochemistry
*Drosophila* pupal wings were prepared for imaging as previously described (*Axelrod, 2001*). Primary antibodies were as follows: mouse anti-Fmi (#74, Developmental Studies Hybridoma Bank), mouse anti-Arm (Developmental Studies Hybridoma Bank), rabbit anti-alpha Tubulin (Abcam, Cambridge, UK), rat anti-tyrosinated Tubulin (Abcam, Cambridge, UK), rat anti-Ds and rat anti-Ft (*Yang et al., 2002*). Images were acquired on a Leica TCS SP5 AOBS confocal microscope using a 63x objective and processed with LAS AF (Leica).

### TEM
For EM analysis, wing imaginal discs and pupal wings were fixed in a mixture of 4% paraformaldehyde and 2% glutaraldehyde in 0.1 M sodium cacodylate buffer, pH 7.3 overnight at RT. Samples were post-fixed in 1% osmium tetroxide in 0.1 M PBS for 1 hr at RT, stained with uranyl acetate,

dehydrated with a graded ethanol series and embedded in EMbed-812 (Electron Microscopy Sciences). Ultrathin sections were cut and analyzed with a JEOL JEM-1400 microscope using a Gatan Orius Camera.

### Time-lapse imaging

For live imaging, pupae were removed from incubation at 25°C 10 min prior to the desired time APF. Pupae were mounted on a small piece of double-sided tape and forceps were used to dissect open a small window in their pupal cases to provide visual access to the live pupal wing. Approximately 50 µl of halocarbon oil was placed over each dissected pupa to allow its release from the tape, following which the pupae were mounted on a VivaScience petriPERM 50 hydrophobic membrane disk in halocarbon oil between pieces of hydrated Whatman paper for in vivo confocal fluorescence microscopy. Videos showing Dsh::GFP and membrane RFP (used to mark wildtype cells) show that internalized Dsh::GFP is always seen to co-stain with RFP. Furthermore, Shimada et al. reported that in fixed images the majority of Dsh and Fz vesicles overlap (*Shimada et al., 2006*). We are therefore confident that internalized Dsh::GFP is in vesicles.

### Image analysis

For analysis of MT orientation we used OrientationJ software (*Rezakhaniha et al., 2012*). We analyzed an average of 20 images from 5 to 10 wing discs or pupal wings for each time point or genotype. The images were taken from appropriate parts of the wing as shown in figures and aligned along the P/D axis (which is plotted as horizontal) of the wing disc or pupal wing.

Analysis of MT anchoring sites in 24 hAPF wings was done using ImageJ.

To analyze the localization of apical tubulin in early third-instar wing pouch we used the cross correlation method as previously described (*Matis et al., 2012*).

For analysis of live imaging time-series, vesicles were only measured and quantified if they were visible in two or more consecutive frames at the level of the adherens junctions. Images were taken at 5 s intervals. We analyzed manually 1192 particle tracks to calculate the net direction of movement (proximal, distal, anterior, posterior or 'stuck'–no change in location between any two consecutive frames). A vector was taken between the first and last points of each track to calculate the net direction of movement.

### Mathematical modeling

A mathematical model, incorporating the proposed mechanism of the Ft/Ds/Fj module to organize MTs, has been created based upon our previously published ODE model (*Ma et al., 2008*). Details are described in *Supplementary file 1*. Simulations of clones (*Figure 5—figure supplement 1*) or of wildtype grids with user defined Fj and Ds gradients (*Figure 5*) were performed to assess kinetics and quasi-steady state levels of polarization.

## Acknowledgements

We thank Seth Blair, David Strutt, Kenneth Irvine, Vladimir Gelfand and the Developmental Studies Hybridoma Bank for reagents; Mike Simon and Axelrod lab members for critical readings of the manuscript. MM was generously supported by a postdoctoral fellowship from the AXA Research Fund. Supported by NIH grants GM059823, GM097081 and P50 GM107615 (J Ferrell PI) to JDA.

## Additional information

### Funding

| Funder | Grant reference number | Author |
| --- | --- | --- |
| National Institutes of Health | GM059823 | Jeffrey D Axelrod |
| National Institutes of Health | GM097081 | Jeffrey D Axelrod |
| National Institutes of Health | GM107615 | Jeffrey D Axelrod |
| Deutsche Forschungsgemeinschaft | EXC1003 | Maja Matis |
| AXA Research Fund | | Maja Matis |

The funders had no role in study design, data collection and interpretation, or the decision to submit the work for publication.

## Author contributions

MM, Conception and design, Acquisition of data, Analysis and interpretation of data, Drafting or revising the article; DAR-G, Acquisition of data, Analysis and interpretation of data; QH, CJT, Wrote the simulation model and performed simulations; JDA, Conception and design, Drafting or revising the article

## Additional files

**Supplementary file**

• Supplementary file 1. Description of modelling

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
