## [Decision Letter]

Thank you for sending your work entitled “Initiation and evolution of asymmetric microtubules by Ft/Ds/Fj in planar polarization of *Drosophila* wing epithelium” for consideration at *eLife*. Your article has been favorably evaluated by K VijayRaghavan (Senior editor) and 4 reviewers, one of whom is a member of our Board of Reviewing Editors.

The Reviewing editor and the other reviewers discussed their comments before we reached this decision, and the Reviewing editor has assembled the following comments to help you prepare a revised submission.

All the reviewers thought there is value in the manuscript, that it adds to the ongoing discussion of the placement of the Ft/Ds and core PCP pathway, and contributes information to the state of microtubules during the development of polarity. However there were 4 major areas that all the reviewers agreed needed to be addressed/improved before this manuscript would be acceptable at *eLife*. In addition there were issues where the figure needed improvement, and the text could benefit from clarification. These points are detailed below.

1) The modeling in its present state does not add substantially to the paper

Reviewer 1: “The authors need to explain some basics about how their model works in the text. How is the Fat-Ds-MT connection modeled? How is it tied to the core proteins? Is there a mechanism for core polarization independent of the MT orientation? As is, the authors state nothing, except for equations buried deep in a supplement.

Then, there is the question of what the model shows, beyond what is intuitively obvious. What the authors claim is oddly spotty. The model is based on a published model that already had the ability to predict autonomy or non-autonomy of the core mutant clones, even without including anything about Fat-Ds-MTs. So the fact that it still does so, the first statement the authors make about the model, is not very surprising.

The old model also showed non-autonomy, so the fact that the new model can show non-autonomy through small fat mutant clones only means that the inclusion of the Fat-Ds-MT equations has not changed much.

The only effect of the new equations that the authors show is that changing between various Ds gradients can change the kinetics of core protein polarization. But they do not even mention the effect of flat Ds expression, or missing Fat. Does it reproduce the effects of missing ds and fj, or uniform ds and fj?

They only show effects on the core proteins. Does the model correctly predict known effects on propagation of Ds-Fat polarization? Does it predict the non-autonomous effects of fat clones on MT orientation?

Reviewer 2: “the model shows how altered Ds gradients might affect MT orientation and Core protein migration. In contrast, the manuscript does not describe experiments in which Ft/Ds gradients alter MT orientation and so determine the direction of Core protein migration. Altered ft/ds activity, MT polarity, PCP protein accumulation and hair polarity”

Reviewer 3: “One of the values of the modelling is that it enabled the authors to test the effects, if any, of alternative Ds gradients. However, none of the alternatives tested resembled the Ds distributions shown in Figure 3—figure supplement 2**.** I would really like to see those tested.”

2) The manuscript needs more extensive quantification and characterization of the MT changes

Reviewer 2: “The most significant finding described in the manuscript (in my opinion) is the ability of altered Ft/Ds activity to be 'instructive' for MT orientation. This is shown in cells surrounding ft loss of function clones. However, in terms of quantification, this is, perhaps, the weakest part of the manuscript. Two clones are shown with just a few cells indicated to show altered MT orientation (Figure 3). The authors made a lot of clones (Figure 3) so a more quantitative analysis of cell non-autonomy could have been undertaken. Also, they might discuss these cell non-autonomous changes in MT orientation with respect to the cell non-autonomous effects of ft clones on wing hair polarity. It seems (to me) that in cells with altered MT orientation (Figure 3), the Core protein Fmi (green) is also localized perpendicular to the clone. I would expect Fz/Dsh to localize with Fmi at one end of the cell. Therefore, it does not appear that Fz/Dsh migration along the reoriented MTs is determining Core protein localization. The authors should discuss this.

“The authors describe a 'strong correlation' between the tubulin staining (presented) which defines MT orientation, and EB1 assay data (not presented), which can show both MT orientation and polarity. Since MT polarity may instruct Core protein migration, it would be useful to see the authors' EB1 data.”

Reviewer 3: “Tyr-tubulin primarily detects very dynamic microtubules. Did the authors see the same bias when using general anti-Tubulin antibodies? If this was stated I did not see it. Given that the microtubule bias they detect is long lasting I am surprised at the antibody choice.”

In the Results section the authors state “…asymmetrical accumulation of tubulin in single “dots” within each cell (Figure 1)”. There are certainly dots that are asymmetric but a substantial minority of the cells have more than one dot. This should be quantitated.

Reviewer 1: “The authors state that “Throughout larval wing discs and pupal wings, MT orientation correlates with the direction of Ds and Fj gradients”. However, at later pupal stages there is a stripe of strong Ds expression that extends into the central portion of the wing blade, which adds a considerable anterior/posterior bias to the Ds gradient in non-central regions of the wing, and which is obvious in the author's photo at 24 hours. Published evidence suggests that this anterior-posterior gradient is instructional for PCP, at least in some mutant backgrounds. Yet the core proteins do not orient along this gradient after their 16 hour reorientation, as noted by the Eaton lab, and from the authors' figures of 24 hour wings the MTs follow the core proteins, not the Ft-Ds-Fj gradient, in non-central regions.

Thus, the correlation is likely less global than the authors state, at least at later stages and non-central regions. If so, the authors are oversimplifying. The pertinent figure, Figure 3—figure supplement 1, shows details of what I suspect are the central region of the wing at 24 hours AP. If the authors have non-central figures that support a global correlation at all stages and locations, they should show them. If not, they need to modify their statements.”

“Disruption of MT organization in the ds mutant was incomplete, displaying unexpected underlying structure, and the altered directionality upon misexpression was not consistent with a simple redirection of MT orientation.”

This is a very interesting result, but the authors should show it. It is also worth a brief discussion, as it differs from the effects of ds fj. How is Fj functioning in a ds mutant, where Ft-Ds binding is lost?

Similarly, in pupal wings MTs are perpendicular to the clone boundary (Figure 3),” I had difficulty seeing this, especially given the variable orientation elsewhere. This would be much more convincing if the authors could quantify it.

3) Clarification of the Ds gradient

Reviewer 1: “the Ds gradient undergoes considerable change from a relatively linear gradient earlier to one with a steep drop and very shallow or flat portion in the 24-30 hour pupal wing (Figure 3—figure supplement 1 and ref. ([27], Hogan et al., 2011a, [25])” and “In the larval wing disc, the Ds gradient is gradual, while in the pupal wing, the gradient of Ds approaches a step gradient as it rearranges first to a projection of high Ds in the central part of the pupal wing, and later to a very high proximal concentrations and a shallow or even flat distal distribution.”

Firstly, it is not clear which of several stages the authors are referring to by “pupal” or “earlier”. Secondly, is the disc gradient really more gradual? The authors do not show a picture, and I do not remember anything convincing from the literature, especially because the proximal half of the wing blade is hiding in a fold at late third. If the authors could show a good picture, that would be valuable, but it would have to include a cross-section to show the tissue in the fold.

The early pupal wing picture in Matakatsu does look very abrupt, with very high Ds near the hinge, but confusingly Figure 3—figure supplement 1 (7 hours) does not look like a step because it cuts off the proximal wing with the highest Ds levels. 1C” (24 hours) looks pretty abrupt proximally, but there looks to be a gradual gradient along the bit that extends out between the central veins.

While the authors make abruptness of the Ds gradient shape a major point of their model, they ignore the modifying effect of the Fj gradient. And the scale of the model is also quite short (only 30 cells) compared with the actual wing blade, so it is not clear how biologically relevant this is.

Finally, what Ds pattern or data does “In comparison to MTs in the distal region, simulation of an unbiased but oriented arrangement is predicted to modestly speed polarization compared to random MTs, but not to change steady state polarization” refer to?

4) Questions on the relevance of the Wnt gradient discussion

Reviewer 2: The authors suggest a second signal from the margin orients microtubules in the non-proximal, central part of the wing and suggest this could be due to Wnts. That is a reasonable suggestion however the paper referenced that showed margin Wnts played a role in PCP made the argument it did this by modulating core protein activity. However, the literature argues the core proteins do not play a role in MT orientation. Are the authors suggesting that margin Wnts have two functions - one to orient MTs and a second to modulate core proteins? Do they think the previous papers missed a disruption of MT orientation in core mutants in the distal region? Of course in a fz or dsh mutant wing the polarity of hairs is only slightly altered in the most distal part of the wing so other factors are likely important.

Reviewer 1: “They also suggest the existence of an additional signal from the wing margin that can orient MTs” and “Distal (peripheral) polarity is independent of Ft function, but appears to depend on a Wnt-dependent signal from the wing margin (45) that one can speculate might orient but not bias MTs.”

First, the authors have no evidence for the source of the missing signals. Second, even if they are wing margin Wnts, there is no evidence that this can control MT polarity. Instead, the quoted work suggests a fairly direct interaction between Wnts and core protein activity, and there is no evidence that changing core protein activity can orient MTs. I think the comments should take this into account. Or the authors could test this by overexpressing Wnt4 and looking.

Reviewer 3: It seems unlikely that Wnt is functioning with Core proteins in a 'non-canonical' pathway as well as acting upstream to control MT orientation.

---

## [Author Response]

1) The modeling in its present state does not add substantially to the paper

We regret that our goal for the modeling was not well articulated, and the revision explains this more clearly. The intent was solely to investigate how changes in the profile of the Ds gradient (in other words, the steepness as a function of distance) along an axis, affect the kinetics of polarization and the steady state polarity. We conclude that differences in the profile have no effect on steady state, and in most cases only subtle effects on the kinetics. This is important, as it seems that the gradients evolve over time, and one might wonder if this has any consequence. We have now extended this result to Fj, with essentially similar results.

We have not attempted to model the radial patterns (a very different question, but we believe this would be a rather uninformative exercise). It would clearly be possible to achieve such a pattern in simulation, but it is not apparent to us what we would learn.

2) The manuscript needs more extensive quantification and characterization of the MT changes

*Reviewer 2: “The most significant finding described in the manuscript (in my opinion) is the ability of altered Ft/Ds activity to be 'instructive' for MT orientation. This is shown in cells surrounding ft loss of function clones. However, in terms of quantification, this is, perhaps, the weakest part of the manuscript. Two clones are shown with just a few cells indicated to show altered MT orientation (*Figure 3*). The authors made a lot of clones (*Figure 3*) so a more quantitative analysis of cell non-autonomy could have been undertaken. Also, they might discuss these cell non-autonomous changes in MT orientation with respect to the cell non-autonomous effects of ft clones on wing hair polarity. It seems (to me) that in cells with altered MT orientation (*Figure 3*), the Core protein Fmi (green) is also localized perpendicular to the clone. I would expect Fz/Dsh to localize with Fmi at one end of the cell. Therefore, it does not appear that Fz/Dsh migration along the reoriented MTs is determining Core protein localization. The authors should discuss this*.

We agree with the reviewer that quantification of MT polarity around Ft clones is warranted. We have now quantified MT orientation around clones and included aggregated results in the new graph in Figure 3. To do this, we analyzed anterior and posterior clone borders, where MTs would normally run parallel to the border. As is evident from the added graph in Figure 3, a majority of MTs is reoriented to run orthogonal to the border.

The reviewer raises a complex question about the cell non-autonomous effects of ft clones on hair polarity. It is indeed correct that some ft clones, primarily larger ones, or those that traverse regions with more irregular cell geometry, such as veins, are more likely to produce swirling hair polarity patterns. This was analyzed previously in Ma et al PNAS 2008. In these cases, Fmi and other core proteins are mislocalized in concordance with the hair polarity pattern. The reviewer wonders why MTs might be oriented differently from the core protein pattern at clone boundaries. Perhaps the simplest case is the one in which a modest sized ft clone does not alter core protein or hair polarity. Here, MTs are still misoriented next to the clone (at A-P oriented edges). To explain this, one must keep in mind that while our hypothesis is that the polarity of a cell is biased by the orientation of MTs, a stronger influence is the tendency of core function to locally align polarity between cells (see [25] for discussion of this phenomenon). To minimize discontinuities in local polarity, this influence will override the influence of the global Ft dependent signal via MTs. A system with this design will produce highly coordinated polarity even in the presence of an imprecise global signal resulting from imprecise gradients, as must necessarily be generated over such a long range.

If the editors believe it would be beneficial, we can include more extensive discussion of this point in the text, but since its essence has already been published and discussed, we feel it would be distracting to the flow of the logic.

*“The authors describe a 'strong correlation' between the tubulin staining (presented) which defines MT orientation, and EB1 assay data (not presented)*, *which can show both MT orientation and polarity. Since MT polarity may instruct Core protein migration, it would be useful to see the authors' EB1 data.”*

A figure and movie of pupal wing co-expressing EB1::GFP and Jupiter::Cherry were added (Figure 1—figure supplement 2).

*Reviewer 3: “Tyr-tubulin primarily detects very dynamic microtubules. Did the authors see the same bias when using general anti-Tubulin antibodies? If this was stated I did not see it. Given that the microtubule bias they detect is long lasting I am surprised at the antibody choice*.*”*

We obtained the same result with both Abs, anti-Tyr Tubulin and anti-alpha Tubulin (Figure 1—figure supplement 2).

*In the Results section the authors state “…asymmetrical accumulation of tubulin in single “dots” within each cell (*Figure 1*).” There are certainly dots that are asymmetric but a substantial minority of the cells have more than one dot*. *This should be quantitated.*

Indeed, a minority of cells has more than one ‘dot’. We speculate that these represent the beginning of the eventual distribution of MT anchoring sites around the cell periphery, though this would be extremely challenging to prove. Furthermore, in cells that are dividing, anti-tubulin Ab also labels the two spindles/centrosomes, and thus in some cases two dots may indicate dividing cells. We have not quantified this, as the number would not have any clear significance.

*Reviewer 1: “The authors state that “Throughout larval wing discs and pupal wings, MT orientation correlates with the direction of Ds and Fj gradients”. However, at later pupal stages there is a stripe of strong Ds expression that extends into the central portion of the wing blade, which adds a considerable anterior/posterior bias to the Ds gradient in non-central regions of the wing, and which is obvious in the author's photo at 24 hours. Published evidence suggests that this anterior-posterior gradient is instructional for PCP, at least in some mutant backgrounds. Yet the core proteins do not orient along this gradient after their 16 hour reorientation, as noted by the Eaton lab, and from the authors' figures of 24 hour wings the MTs follow the core proteins, not the Ft-Ds-Fj gradient, in non-central regions*.

In early pupal wings (7 hAPF), the Ds gradient is radial, but gradually evolves into a more linear organization. We agree with the reviewer that there is still a detectable stripe of Ds expression near the proximal part of the L3 vein at 24 hours, but careful examination reveals that this impression, gleaned from lower magnification images, reflects higher cytosolic concentrations of Ds and also perhaps the small cells that comprise the vein. In contrast, membrane bound Ds, the relevant population, is polarized perpendicular to this Ds gradient along the P/D axis by 24 h APF. We now include additional images showing asymmetry of membrane associated Ds along the P/D axis in this region of the wing (Figure 3—figure supplement 4).

*Thus, the correlation is likely less global than the authors state, at least at later stages and non-central regions. If so, the authors are oversimplifying. The pertinent figure,*
Figure 3—figure supplement 1*, shows details of what I suspect are the central region of the wing at 24 hours AP. If the authors have non-central figures that support a global correlation at all stages and locations, they should show them. If not, they need to modify their statements*.*”*

We now include additional images of the whole wing showing Ds and core polarity at 26APF, as well as higher magnification images from selected regions. However, in a different context, the reviewer is correct: We point out that there is a possible discrepancy in the orientation of Ds with respect to core protein orientation in the posterior region of the wing, near the posterior margin. In this region, the core proteins appear to be oriented with a mostly proximal-distal polarity, whereas the Ds stain appears to be oriented with a somewhat A-P polarity. While this potential discrepancy may be enhanced by ‘illusion’ due to the relatively elongated cell shapes and cell packing organization that is observed in this region, we think there is indeed some difference in direction in this region. In high magnification images (some more than others), there is modest evident enrichment of Ds in a more posterior orientation. We do not have a solid explanation, but now discuss several possibilities in the text. It is important to note that this is a region of the wing where the Ft/Ds/Fj module is less important, or at least perhaps redundant with another signal.

*“Disruption of MT organization in the ds mutant was incomplete, displaying unexpected underlying structure, and the altered directionality upon misexpression was not consistent with a simple redirection of MT orientation*.*”*

*This is a very interesting result, but the authors should show it. It is also worth a brief discussion, as it differs from the effects of ds fj*.

The first part of this sentence is meant to refer to results in Harumoto (2010), where in the distal part of the wing, MTs are P-D without much or any bias in polarity in wild type, but in the ds mutant, show a modest plus end proximal polarity. We have no good explanation for this, but it may be related to the observation that Ft clearly has some influence in the absence of Ds, or perhaps to other influences that are unmasked by loss of Ds. The second part refers to a complex experiment in which MT polarity was measured across the boundary of en-GAL4 UAS-Ds expression, and gave a confusing result. Both of these allusions were unnecessarily oblique, and have been deleted.

We don’t think there’s an important difference between the Harumoto ds loss of function result and our Ds Fj analysis, as these statements refer to different locations. Their result to which we refer was near the distal end of the wing; they also show loss of organization proximally.

Similarly, our results show loss of parallel organization in the proximal/central part of the wing but not distally (with some other signal presumably organizing the distal region, as discussed in this manuscript and below).

*How is Fj functioning in a ds mutant*, *where Ft-Ds binding is lost?*

Again, we don’t know, but may be related to the Ds independent function of Ft, which is also a substrate for Fj. This is an interesting and important question that is well beyond the scope of this manuscript.

*Similarly, in pupal wings MTs are perpendicular to the clone boundary (*Figure 3*),” I had difficulty seeing this, especially given the variable orientation elsewhere. This would be much more convincing if the authors could quantify it*.

As discussed above, this is now quantified.

3) Clarification of the Ds gradient

*Reviewer 1: “the Ds gradient undergoes considerable change from a relatively linear gradient earlier to one with a steep drop and very shallow or flat portion in the 24-30 hour pupal wing (*Figure 3—figure supplement 1
*and ref. (*[27]*, Hogan et al., 2011a,*
[25]*)” and “In the larval wing disc, the Ds gradient is gradual, while in the pupal wing, the gradient of Ds approaches a step gradient as it rearranges first to a projection of high Ds in the central part of the pupal wing, and later to a very high proximal concentrations and a shallow or even flat distal distribution*.*”*

*Firstly, it is not clear which of several stages the authors are referring to by “pupal” or “earlier”*.

Corrected – Figures of several stages have been added as supplements to Figure 3. We have tried to clarify the language used to describe these observations.

*Secondly, is the disc gradient really more gradual? The authors do not show a picture, and I do not remember anything convincing from the literature, especially because the proximal half of the wing blade is hiding in a fold at late third. If the authors could show a good picture, that would be valuable, but it would have to include a cross-section to show the tissue in the fold*.

We have added an image showing this, at least for the distal part of the wing blade that is not hidden in the fold (Figure 3—figure supplement 1).

*The early pupal wing picture in Matakatsu does look very abrupt, with very high Ds near the hinge, but confusingly*
Figure 3—figure supplement 1
*(7 hours) does not look like a step because it cuts off the proximal wing with the highest Ds levels*.

While our image of a 7 hour wing does not show all of the hinge, it was visualized by Matakatsu, (although with LacZ, so that staining may reflect earlier expression).

*1C“ (24 hours) looks pretty abrupt proximally, but there looks to be a gradual gradient along the bit that extends out between the central veins*.

Indeed, the gradient appears to be steep at 24 hours, with a shallower portion extending into the blade. However, as discussed above, and now in the text as well, all of these observations from low magnification images should be taken with a grain of salt, as they do not distinguish membrane from cytoplasmic/internal protein, nor can they distinguish true differences in amount from variation in cell size (smaller cells give the appearance of higher concentrations at low mag). To address all of these concerns, we have added some explanation to the text, and pointed out the important take home message, which is that the true steepness of the gradient of membrane associated Ds is ambiguous and might change over time. This should be distinguished from the direction of Ds asymmetry, which can be clearly read from high magnification images.

As a final note to this point, there is now evidence that non-autonomy and local feedback within the Ft-Ds mechanism that would be expected to locally align the polarity of these molecules, perhaps overriding the effects of reading gradients alone.

*While the authors make abruptness of the Ds gradient shape a major point of their model, they ignore the modifying effect of the Fj gradient. And the scale of the model is also quite short (only 30 cells) compared with the actual wing blade, so it is not clear how biologically relevant this is*.

We chose to address only the Ds gradient because this is the one for which there is any real data. Because of the lack of a Fj antibody that works for immunofluorescence, there are only LacZ data for Fj, and perdurance makes these unreliable for addressing changes with time. We therefore left the Fj gradient constant. However, we have now done similar simulations for Fj, and find essentially similar results. This is now noted in the text.

*Finally, what Ds pattern or data does “In comparison to MTs in the distal region, simulation of an unbiased but oriented arrangement is predicted to modestly speed polarization compared to random MTs*, *but not to change steady state polarization” refer to?*

This experiment is meant to test the potential relevance of proximally-distally oriented but unbiased MTs that is actually observed in the distal part of the wing as reported by [18]. The result is a modest increase in the rate of polarization, but equivalent steady state.

This is part of the larger question, much debated in our lab, about the relevance of these MTs in the distal part of the wing. The language in the manuscript has been clarified.

*4) Questions on the relevance of*
*the Wnt gradient discussion*

*[…] Reviewer 1: “They also suggest the existence of an additional signal from the wing margin that can orient MTs” and “Distal (peripheral) polarity is independent of Ft function, but appears to depend on a Wnt-dependent signal from the wing margin (*[45]*) that one can speculate might orient but not bias MTs*.*”*

*First, the authors have no evidence for the source of the missing signals. Second, even if they are wing margin Wnts, there is no evidence that this can control MT polarity. Instead, the quoted work suggests a fairly direct interaction between Wnts and core protein activity, and there is no evidence that changing core protein activity can orient MTs. I think the comments should take this into account. Or the authors could test this by overexpressing Wnt4 and looking*.

*Reviewer 3: It seems unlikely that Wnt is functioning with Core proteins in a 'non-canonical' pathway as well as acting upstream to control MT orientation*.

Wu et al showed that Wnt 4 and Wg (or at least Wnt4 plus some other Wg dependent signal) influence polarity in the distal part of the wing. However, the data on which they base their model that this occurs by affecting Fz-Vang interaction should be taken with a grain of salt, as the assay depended on a cell culture model with high concentrations of added Wnt, and the physiological relevance is therefore unclear. Furthermore, their model would require that concentrations of ligand are sufficiently different across a single cell diameter to influence polarity outcomes, also a problem of questionable plausibility. Therefore, we feel it is valuable to consider other possibilities. And, of course, yet other signals may be at play.

With respect to Wnt4, we do indeed see an ability to modify MT orientation, and would therefore like to at least suggest the possibility of a different mechanism. This alternative would not necessarily be mutually exclusive, though it might stretch plausibility to suggest that Wnts work by two distinct mechanisms.

We have included the data for this, and elaborated some in the discussion, while attempting to minimize unnecessarily controversial assertions.